# The Early Silurian Sedimentary Environment of Middle-Upper Yangtze: Lithological and Palaeontological Evidence and Impact on Shale Gas Reservoir

**Xiaorong Qu [1,2,3], Yanming Zhu [1,2,*], Yang Wang [1,2] and Fuhua Shang [1,2]**

1   Key Laboratory of CBM Resource and Reservoir Formation Course, Ministry of Education, China University of Mining and Technology, Xuzhou 221006, China
2   School of Resources and Geoscience, China University of Mining and Technology, Xuzhou 221006, China
3   Shanxi Provincial Coal Geological Exploration Research Institute, Taiyuan 030000, China
*   Correspondence: ymzhucumt@126.com; Tel.: +86-0516-83591012

**Abstract:** The organic-enriched thick shale at the bottom of Longmaxi Formation is laterally continuous distributed and has been proven to be of good production capability in Fuling of Upper Yangtze. Uplifts that developed during the sedimentation influenced the reservoir characteristics by taking control of the sedimentary environment and provenance. The sedimentary environments are mainly deep-water shelf, shallow-water shelf, and tidal flat. By analyzing reservoir characteristic of these three environments, the deep-water shelf, which dominated the early stage of sedimentation, formed a high-quality reservoir with high TOC (Total Organic Carbon) content, porosity, and brittleness, while the environment was maintained around the basin centre until the Early Silurian. The shales deposited under the shallow-water environment were of low porosity because of the increasing calcareous and argillaceous contents. Sediments which formed on the tidal flat were arenaceous and of the lowest TOC content as the organic preservation conditions deteriorated. The good correlation of graptolite abundance and TOC content, and high porosity within graptolite fossils emphasize the importance of palaeontological development. The argillaceous cap over the Longmaxi shale is of good sealing capability, and the continuous sedimentation zone along southern Sichuan–eastern Chongqing is the best optimized hydrocarbon-bearing system. However, a weak interface on the discontinuity is the potential lateral pathway for gas diffusion at Northern Guizhou and Western Hunan, but on the southeast margin where the dark shale and the tidal sandstone contact, it promises to form a tight gas reservoir.

**Keywords:** graptolite; biozonation; lithological characteristics; Middle-Upper Yangtze; Longmaxi Formation

## 1. Introduction

The late Ordovician–early Silurian Longmaxi Formation in the Middle–Upper Yangtze is a laterally continuous stratum seen as a hydrocarbon source, and now proven to be productive in Fuling of Upper Yangtze, which is now under commercial exploration with a gas flow of $(15.3–155.8) \times 10^4$ m³/day [1]. The lower member of Longmaxi Formation formed under deep shelf environment, along with euxinic and reductive water background, appropriate sedimentary rateis of high hydrocarbon generation potential [2–4]. The hydrocarbon source rock quality of the upper member is generally deteriorating as tectonic transformations leading to a shallower basin [5–7]. The typical Longmaxi shale gas reservoirs in Upper Yangtze were well studied [6,8]. The zone of the highest potential is the lower member, with

high TOC (Total Organic Carbon) content and brittleness, as well as high porosity and gas capacity. The organic-rich shale formed a laterally continuous, high-thickness zone from southern Sichuan to eastern Chongqing, with a maximum thickness of 200 m [9]. The coupling of the sedimentary environment and a high quality reservoir indicates the importance of sedimentary environment study during reservoir evaluation [10]. The pore structure characteristic is another key parameter in shale gas evaluation, that is under the control of provenance, which is also influenced by the sedimentary environment.

Traditionally the studies of the sedimentary environment are based on lithology, logging, geochemistry, palaeontology [11,12]. However, some research in shale studies pay more attention on the former but put less weight on palaeontology which can be an important factor as well. There are detailed studies about the Ordovician and Silurian strata in Middle–Upper Yangtze since the last century, while these studies mainly focused on biostratigraphy and chronostratigraphy [13–16]. These studies provide detailed and accurate materials for our analysis. By clarify the distribution of each graptolite zonation, the boundary of the sedimentary zone can be bounded. Furthermore, palaeontology would provide more details than merely landscape; for instance, the water condition and the depth of the living species. The discontinuous contact between Ordovician and Silurian was observed in the early studies on fossil zonation, which indicated the distinctive sea level and tectonic reconstructions during the sedimentation of Longmaxi Foramtion throughout the Middle–Upper Yangtze.

Graptolite is the most representative fossil in Longmaxi Formation. Beside graptolite, there are few reports about foraminifer, chitinozoa and microflora [17]. Most of them are consumers rather than producers in the food chain. Due to the euxinic and reductive water background, most of the producers (plants) were decomposed while graptolites were preserved as carbonized laminae. What is more important, graptolite work as the consumers in the food chain, so that the abundance of graptolites is a good indication of the abundance of the producers, which is the main source of hydrocarbon generation. Hence the graptolite abundance is well correlated with the TOC content.

The research area can be divided into four regions (Figure 1), the continuously deposited basin centre in Upper Yangtze, and three zones influenced by uplifts, the Hanzhong Uplift (northern Upper Yangtze), Qianzhong Uplift (northern Guizhou, Upper Yangtze), and on the east, the Yichang submarine highland (west of Hubei, Middle Yangtze). These uplifts were the result of tectonic reconstruction and influenced not only the landscape but also the sedimentary environment during deposition, and furthermore led to varied contact relations of Longmaxi Formation and underlying strata in the four regions, as well as lithologies association both vertically and laterally. There are irony or argillaceous weathering crust in northern Guizhou and western Hubei indicating the discontinuity, while in other parts of the basin the discontinuity is only identified by palaeontology when closer to the basin centre. Hence, zonation is helpful in detailed studies of the dynamic uplift development. Along with the sedimentation and lithologies in the Middle–Upper Yangtze, the control mechanism of the sedimentary environment of the organic-rich shale development and the influence of gas accumulation are further revealed.

## 2. Investigation and Methodology

The Yangtze fossil zonation of the Ordovician and Silurian extinctions has been well studied, and the zonation is slightly different in the study area (Figure 1). This paper discusses the zonation by referencing the Stratigraphic Chart of China [18] to compare the zonation in different regions. Chen Xu et al. [19] designed a classification based on graptolite zonation for Wufeng Formation and Longmaxi Formation (WF1-WF5, LM1-LM9), which is applicable and the zonation can be well compared with the Stratigraphic Chart.

By studying 13 outcrops or wells, graptolite development, lithologies, mineral composition, and reservoir characteristics were analyzed. The distance between samples for the lower member is 0.5 m, and the meantime make sure there is at least one sample in each zonation as the thicknesses of the earlier zonations of Longmaxi Formation are usually less than 1m, and for the upper member with great thickness (50–200 m), the distances increased to 1 or 2 m. The palaeontology development and

lithologies were identified in the field and microscopically observed back in lab, mineral composition and pore structure characteristics were tested by XRD, mercury intrusion and liquid nitrogen adsorption, and observed by SEM, respectively. The XRD experiment was conducted on Rigaku Ultima IV using samples crushed to 200–300 mesh. The Brittle Index is identified as the content of quartz over the summation of quartz, clay, and carbonate. The mercury intrusion was conducted on Micromeritics 9510, and liquid nitrogen adsorption on Micromeritics ASAP 2020 (Micromeritics, Norcross, GA, USA) at 77K. There were both focused-ion-beam polished and unpolished samples imaged by field emission SEM.

| Stage | Biozonation | Formation | Sanxia, Hubei [18] | Ningqiang, Shaanxi [19] | Qijiang, Chongqing [20] | Tongzi, Guizhou [21] | Chengkou, Chongqing [22] | Xingwen, Sichuan [23] | Yangtze [17] |
|---|---|---|---|---|---|---|---|---|---|
| Telychian | *Spirograptus guerichi* | | *Monoclimacis arcuata* | *Streptograptus nodifer* | *Monograptus atucata* | | *Rastrites maximus* / *Petalolithus folium* | | LM9/NJ1 (*Spirogra. guerichi*) |
| Aeronian | *Stimulograptus sedgwickii* | Longmaxi Fm | *Stimulograptus sedgwickii* | *Monograptus sedgwickii* | *Monograptus sedgwickii* | | *Pristiograptus leptotheca* | *Stimulograptus sedgwickii* | LM8 (*Stimulogr. sedgwickii*) |
| Aeronian | *Lituigraptus convolutus* | Longmaxi Fm | *Lituigraptus convolutus* | *Demirastrites convolutus* | *Oktavites communis* | *Oktavites communis* | *Pristiograptus leptotheca* | *Lituigraptus convolutus* | LM7 (*Lituigraptus convolutus*) |
| Aeronian | *Demiras. pectinatus- Monogr. argenteus* | Longmaxi Fm | *Monogr. aregenteus* / *Neo. magnus- Neo. thuringiacus* | ? | *Oktavites communis* | *Oktavites communis* | *Pristiograptus leptotheca* | *Demirastrites triangulatus* | LM6 (*Demirastrites triangulatus*) |
| Rhuddanian | *Demirastrites triangulatus* | Longmaxi Fm | *Demirastrites triangulatus* | *Coronograptus cyphus – Monoclimacis lunata* | *Demirastrites triangulatus* | *Coronogr. gregarius*: *Demirastrites triangulatus* / *Demirastrites triangulatus* | *Pristiograptus leei* | *Demirastrites triangulatus* | LM6 (*Demirastrites triangulatus*) |
| Rhuddanian | *Coronograptus cyphus* | Longmaxi Fm | *Coronograptus cyphus* / *Huttagraptus acinaces* | (?) | *Pristiograptus cyphus - Pr. leei* | *Pristiogr. cyphus - Momoclimacis lunata* | *Pristiograptus leei* | *Coronograptus cyphus* | LM5 (*Coronogr. cyphus*) |
| Rhuddanian | *Cystograptus vesiculosus* | Longmaxi Fm | *Cystograptus vesiculosus* | ? | *Orthograptus vesiculosus* | *Orthograptus vesiculosus* | *Orthograptus vesiculosus* | *Cystograptus vesiculosus* | LM4 (*Cystogr. vesiculosus*) |
| Rhuddanian | *Parakidograptus acumininatus* | Longmaxi Fm | *Parakidograptus acumininatus* | | *Akidograptus ascensus* | | *Parakidograptus acumininatus* | *Parakidograptus acumininatus* | LM3 (*Parakidogr. acumininatus*) |
| Hirnantian | *Akidograptus ascensus* | Longmaxi Fm | *Normalograptus persculptus* | *Climacograptus miserablis* | *Akidograptus ascensus* | *Akidograptus ascensus* | *Parakidograptus acumininatus* | *Akidograptus ascensus* | LM2 (*Akidogr. ascensus*) |
| Hirnantian | *Normalograptus persculptus* | Wufeng Fm | | Fossil absent | *Glyptograptus persculptus* | *Glyptogr. persculptus - G. sinuatus* | *Diplogr. modestus - Glyptogr. gracilis* | *Metabolograptus persculptus* | LM1 (*Persculptogr. persculptus*) |
| Hirnantian | *Normalograptus extraodinarius* | Wufeng Fm | *Hirnantia* fauna | *Hirnantia* Fauna | *Hirnantia* Fauna | *Hirnantia* Fauna | | *Hirnantia* Fauna | WF4 (*Normalogr. extraodinarius*) |

1- Kuanyinqiao Member; 2- Wulipo Member

**Figure 1.** Stratigraphic framework and graptolite zonation in Middle–Upper Yangtze.

The outcrops and wells investigated mainly distributed around the basin centre. Abundant studies would provide more detailed information about the graptolite zonation that would help to reveal the changing landscape and lithologies. However, there is no accurate information about the TOC content but only the description of color that gives a clue about the TOC content, thus colors darker than dark grey were identified as promising members. Correspondingly TOC contents higher than 1% were taken as favorable members as well because of the dark shade.

Two profiles, the north (53-44-11-12-14-XIII-VII) and south (I-33-30-41-20-18-40-VIII-48), were selected to represent the graptolite development and lithological association in the four target regions to illustrate the sedimentary environment change (Figure 2). These two profiles are representative in elucidating the change of environment as the profiles run along the margins of the uplifts involved.

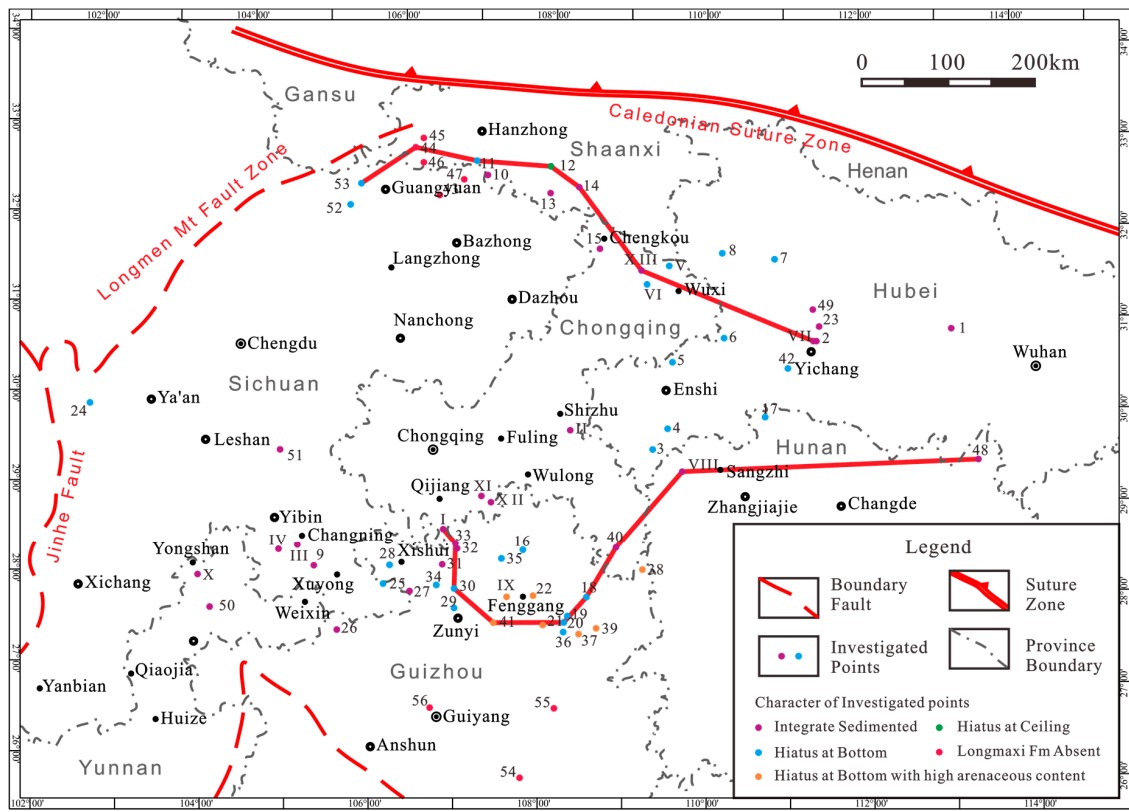

**Figure 2.** Distribution of investigated and reference points.Investigated points: I—Guanyinqiao, Qijiang, II—Dafeng'ao, Shizhu, III—Changning, Yibin, IV—Gongxian, Yibin, V—Bailu, Wuxi, VI—Tianba, Wuxi, VII—Fenxiang, Yichang, VIII—Bixi, Longshan, IX—Tianping, Fenggang, X—Yinjiawan, Yongshan, XI—Sanquan, Nanchuan, XII—Dayou, Nanchuan, XIII—Well WX2. Reference points: 1—Daozimiao, Jingshan [20]; 2—Wangjiawan, Yichang [20]; 3—Laifeng, Sanbaoling [20]; 4—Gaoluo, Xuan'en [20]; 5—Taiyanghe Enshi [21]; 6—Siyangqiao, Badong [20]; 7—Qingquan, Shennongjia [20]; 8—Laomatou, Zhushan [20]; 9—Qilinxiang, Xingwen [22]; 10—Fucheng, Nanzheng [23]; 11—Xihe, Nanzheng [23]; 12—Sanlangpu, Xixiang [23]; 13—Xingzishan, Zhenba [23]; 14—Zihuang, Ziyang [23]; 15—Datangkou, Chengkou [23]; 16—Longjingpo, Wuchuan [14]; 17—Longchihe, Shimen [14]; 18—Heshui, Yinjiang [14]; 19—Sigou, Shiqian [14]; 20—Leijiatun, Shiqian [24]; 21—Wenjiadian Sinan [14]; 22—Dongkala, Fenggang [14]; 23—Dazhongba, Yibin [14]; 24—Xinglong, Luding [25]; 25—Taiping, Gulin [15]; 26—Yanzikou, Bijie [15]; 27—Yangliugou, Huairen [15]; 28—Sanhuai, Xishui [15]; 29—Shizipu, Zunyi [15]; 30—Banqiao, Zunyi [26]; 31—Songkan, Tongzi [15]; 32—Liangfengya, Tongzi [15]; 33—Hanjiadian Tongzi [24]; 34—Liaoyuan, Tongzi [15]; 35—Ganxi, Yanhe [15]; 36—North of Shiqian [15]; 37—Zhoujiaba, Yinjiang [15]; 38—Ludiping Songtao [15]; 39—Huoshaoqiao, Guizhou [15]; 40—Rongxi, Xiushan [14]; 41—Niuchang, Meitan [27]; 42—Changyang, Yinchang [16]; 43—Dalianghui, Wangcang [23]; 44—Houjiayuan, Ningqiang [23]; 45—Zhaojiaba, Ningqiang [23]; 46—Maopinggou, Ningqiang [23]; 47—Qiaoting, Nanjiang [23];48—Linxiang, Hunan [28]; 49—Goujiaya, Yuan'an [29]; 50—Huanggexi, Daguan [28]; 51—Weiyuan, Sichuan [28]; 52—Changjianggou, Guangyuan [28]; 53—Qilixiang, Guangyuan [28]; 54—Dahe, Dujun [30]; 55—Shuizhu, Kaili [31]; 56—Wudang, Guiyang [32].

## 3. Fossils and Lithological Characteristics

### 3.1. Hanzhong Uplift, North of the Upper Yangtze

The Longmaxi Formation outcrops are well developed north of the Upper Yangtze, and the thickness of this formation varies from tens of meters to more than a hundred meters. The lithology is mainly silty shale and argillaceous silt, and the distribution of black carbonaceous shale is laterally continuous on the bottom of the formation. The frequent crust oscillation (called Xixiang Uplift)

caused by the Qinling-Longmenshan Fault controlled the sedimentation of this region and led to the discontinuity between Ordovician and Silurian strata. The profiles along north of the Yangtze show that the ceiling of missing strata arises from the southwest to northeast under the dual influence of Xixiang Uplift and sea level oscillation.

By comparing the profiles (Figures 3 and 4), there are more missing strata on the west. For example, Point 52, where the 4 m Longmaxi Formation extends through five of nine zonations (*ascensus-trangulatus*), and Longmaxi Formation at Point 53, which is mainly silt and fine sandstone without dark shale formed since Areonian *pectinatus*, and lying over the Ordorvician Pagoda Formation directly with a great thickness (134.4 m). The zonation missing is even higher on the north, and the whole Longmaxi Formation can be absent around Guangyuan, for example Point 44 and 47 (Figure 4). Near Hannan archicontinent, only the bottom of Longmaxi Formation is preserved at Point 12, and the underlying Wufeng Formation is pebbly sandstone, showing the near-shore rapid deposition. The discontinuity disappears on the southeast, and the thickness of dark shale and the formation increases remarkably. On the southeast of point 12, there is Rhuddanian dark shale sedimentation again with increasing thickness (14-XIII), and tracking to the southeast (Point VII), it is under the influence of uplift in the Middle Yangtze.

In the late Aeronian, most of the north Yangtze started to accept depositions, except the Guangyuan region and Hannan archicontiment, and the sediments are mainly yellow-green arenaceous shale without dark shale. By taking the Rhuddanian sediment into consideration, it can be speculated that these uplifts already existed since Hirnantian and narrowed down in the Rhuddanian.

The north margin was strongly reconstructed by the orogenic belt, so as the basin landscape. The missing upper member of Point 12 indicated rapid uplift of the archicontinent, while for Point 10 and 13 in the northern Yangtze, the dark shale sustained until Nanjiang Formaiton of Telychian, indicating that the basin was partially deepened because of the tectonic compression.

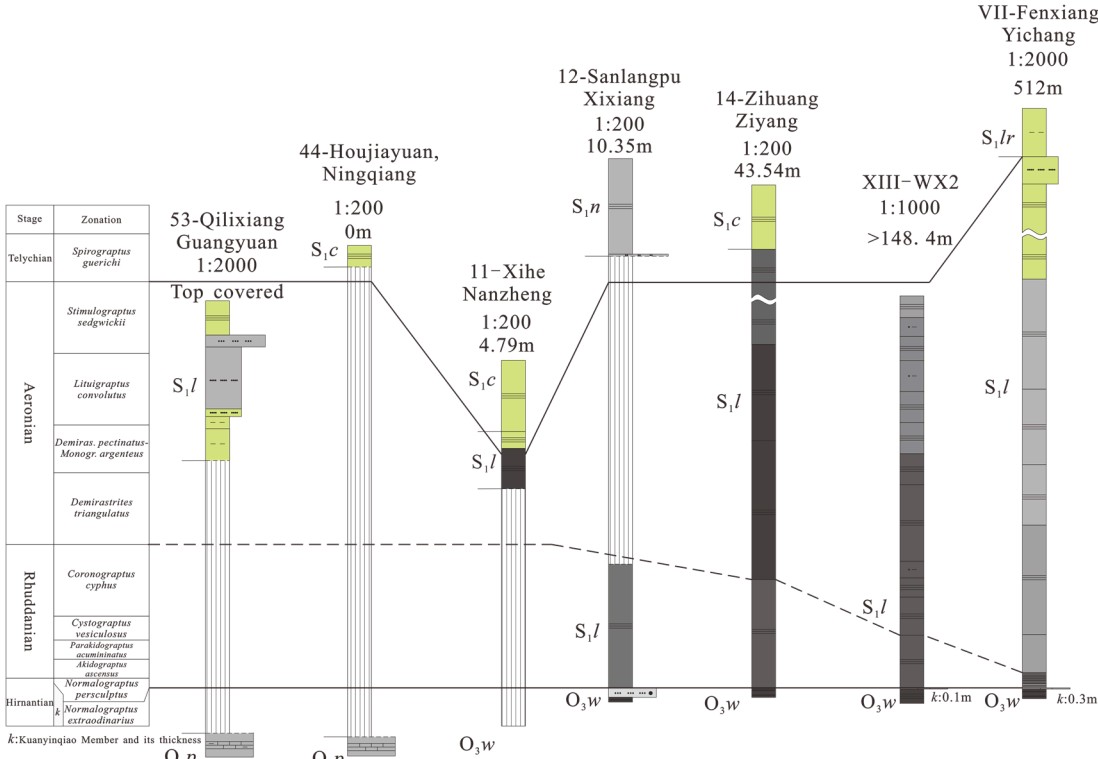

**Figure 3.** Development of Longmaxi Formation on the north of Middle–Upper Yangtze.

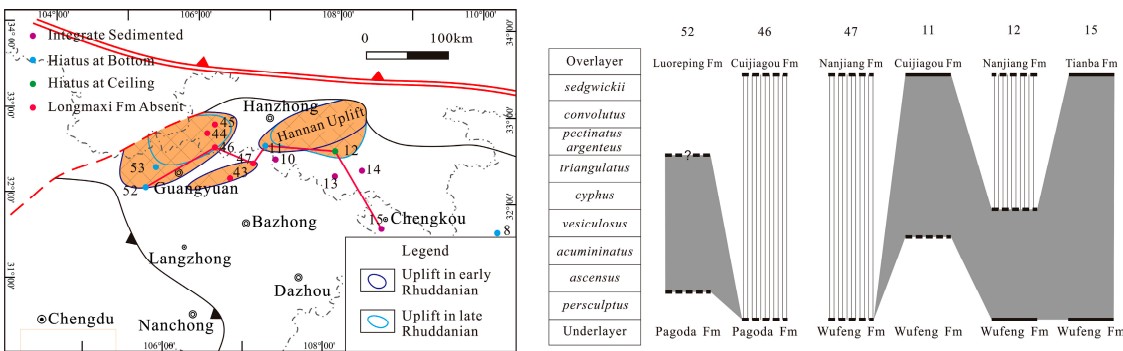

**Figure 4.** The changing archicontinent and uplifts in north Yangtze.

## 3.2. Qianzhong Uplift, Upper Yangtze

The lithological characteristic and fossil zonation analysis along the southern Yangtze clearly represents the transition of the Qianzhong uplift (Figure 5). Wudang, Guiyang (Point 56) lies on the south of the Qianzhong uplift (Figure 6), and late Aeronian Gaotianzhai Formation, whose lithology is fuchsia calcareous-breccia silt, overlies the middle Ordovician Huanghuachang Formation directly without early Silurian sediments [32,33]. North of Point 56, it was brown-yellow silty shale and gray shale, for instance Meitan, Sinan (Point 41), and the fossils were mainly branchiopod with few graptolites, which was quite different from the Longmaxi Formation. This indicated normal shallow water near the shore, and the formation was named the Niuchang Formation after detailed studies, isochronous to deposition during the Rhuddanian C. vesiculosus zonation to Aeronian *D. triangulates* [27,34]. On the east of Northern Guizhou, zonation in Wufeng and the bottom of Longmaxi was missing at Point 20 and 21, while the Kuanyinqiao limestone was well developed, and at Point 21 there is irony weathering crust on the Kuanyinqiao limestone [15,35]. The weathering crust also existed at Point XI, and the thickest Kuanyinqiao limestone (11 m) was reported near Point XI [14]. To the northwest, the missing zonation narrowed down, and at Point 25 and 28 only the last zonation of Hinantian (*N. persculptus* zonation) was missing.

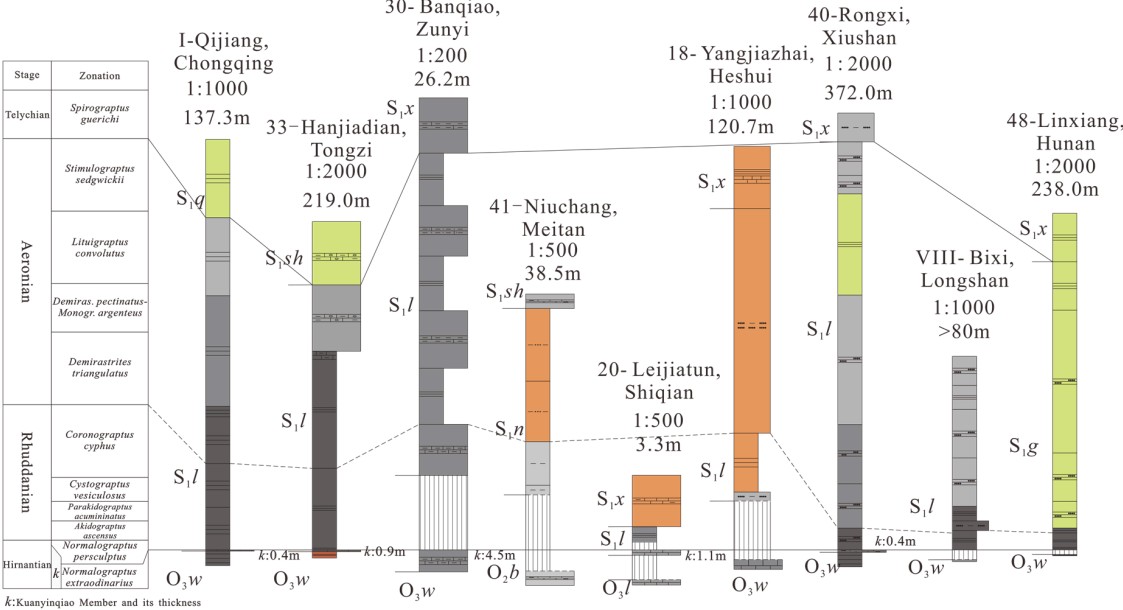

**Figure 5.** Lithologies and fossil development of Longmaxi Formation along the south of Yangtze.

Northeast Guizhou and Southeast Chongqing lie between the Qianzhong Uplift and the Yichang Uplift in Middle Yangtze. The earliest two zonation at the bottom of Aeronian at Point 18 was

absent [14], while Point 40 on the northeast was continuously sedimented, and the *N. persculptus* was missing again on the northeast, as it is under the influence of uplift in the Middle Yangtze. North of Point 40, the sediments were much thicker approaching the basin centre, and the dark shale can be more than 100 m at well Yuye 1, Pengshui [36].

After the glaciation of the late Ordovician, the sea level started to rise subsequentially. However, the Qianzhong Uplift kept expanding to the north (Figure 6a), indicating a high uplifting rate since the late Hinantian. The uplift zone showed a complicated landscape. The benthonic fauna with limestone near the southern margin indicated a normal water environment, while the plankton graptolite with horizontal bedding siliceous or carbonaceous shale offshore demonstrated a euxinic, reductive environment. The rising sea level submerging the uplift led to the withdraw of the seashore to the south during the late Rhuddanian, and started to receive sediments. The expanding sea invaded south and formed the shallow channel between Qianzhong and Xuefeng Uplift in Aeronian, which was filled with sandy sediments. The sediment around the basin centre was mainly carbonate or silty shales(Figure 6b).

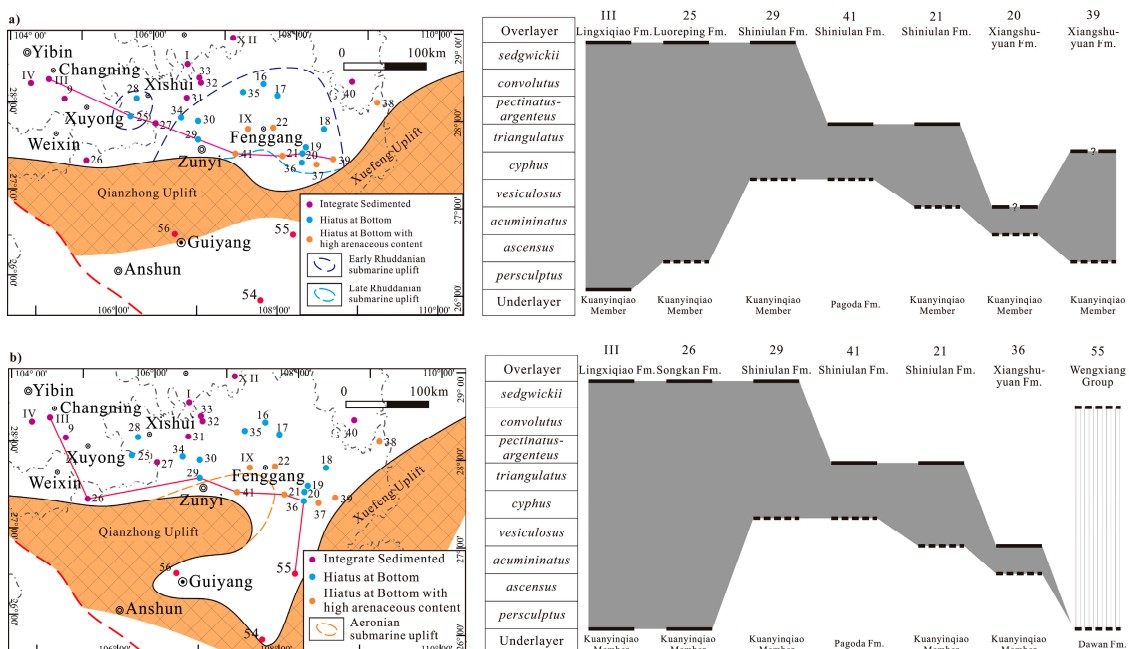

**Figure 6.** Change of Qianzhong Uplift on the southern Yangtze (**a**) Rhuddanian, (**b**) Aeronian.

### 3.3. Yichang Submarine Highland, Middle Yangtze

The uplift in the Middle Yangtze is called Yichang Uplift and lies to the west of Hunan and Hubei. The profile along the margin of the uplift shows the strata development clearly (Figure 7). The diachroneity of the Longmaxi Formation bottom is apparent, and the higher zonation absent, the higher the base of dark shale would be. The influence of uplift was sustained until *C. vesiculosus* zonation (Point 5, 7). Point VII and 23 turned to continuous contact on the east.

The investigation of Bixi, Longshan (Point VIII) demonstrated the great thickness of the Longmaxi Formation (larger than 80 m) with the dark shale (14 m) concentrated at the bottom (Figure 5). The zonation of the late Hirnantian at Point VIII was absent, indicating that the uplift raised and submerged early in this region. On the very east of the target area, Point 1 and 48 represent similar character, with integrated Longmaxi Formation but a break in the Wufeng Formation. It can be inferred that the uplift existed already in the early Hirnantian, and then shrank to the west quickly during the late Hirnantian under the influence of the rising sea level, while in the late Rhuddanian, the uplift was wholly submerged, with dark shales formed at the bottom.

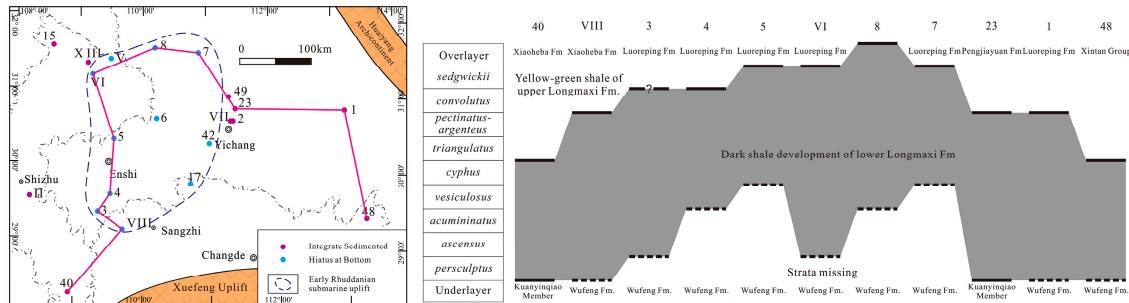

**Figure 7.** Transition of Yichang Uplift west of Hunan and Hubei.

## 4. Distribution of Sedimentary Environments and Lithologies

### 4.1. Rhuddanian

During the sedimentation of Wufeng Formation, under the influence of glaciation, sea levels descended to 60–80 m inferred by Hirnantian fauna [37,38]. A large area of submarine uplift was exposed, including Zunyi–Songtao of Guizhou, Gulin–Xishui in the north of Guizhou Province, and west of Hunan–Hubei. Subsequentially, the melting glaciers caused a quick rise in sea level since the *N. persculptus* zonation in late Hirnantian, and the basin centre was 200–400 m deep, equivalent to the depth before Kuanyinqiao deposited [38]. The water was euxinic and reductive [38,39], forming a wide-spread organic-enriched dark shale with simple parallel bedding. The uplifts existing in the early Hirnantian shrank over a large range or even disappeared, as at Jingshan (Point 1)–Linxiang (Point 48) in the east of Middle Yangtze and Guangyuan (Point 53) in the north Upper Yangtze, and started to accept argillaceous sediments, thus the dark shale is rather thin. Delta-dominated sediments are well developed along the margin of the southern archicontinents [40,41].

Uplifts in northern Yangtze also shrank under the influence of rising sea level and the lithofacies were dominated by sandy deposition, while the Hannan Archicontinent expanded (Figure 8). The thicknesses of both the Longmaxi Formation and dark shales were rather thin around the uplifts (for Point 11, the thickness of dark shale is 3.38 m while the formation is 4.79 m), but increased remarkably around Chengkou, Wuxi (Point XIII) on the east, indicating the environment shifted from shallow water to lentic bathyal environment.

Despite the rising sea level, the early zonation of Longmaxi Formation was absent at north Guizhou as the Qianzhong Uplift continued expanding to the north, along the interface forming a weathering crust or a silty bottom. On the south margin, the Niuchang Formation was formed, indicating an oxygenated water background. The basin depth increased gradually from the Qianzhong Uplift to the northern basin centre, forming a deep-water shelf, and the thickness of dark shale deposited increased to the north. The dark shale of the lower member in the Longmaxi Formation was siliceous or carbonaceous, with little silt and enriched graptolite fossil preserved along the parallel bedding, indicating a lentic water background. During the late Rhuddanian, as the uplift rate decreased and the sea level continued rising, the coastline withdrew to the south.

### 4.2. Aeronian

The sea level started to drop in the late Rhuddanian [26,38], and most of the Yangtze region turned to shallow water in the Aeronian. This led to a series of changes. The sedimentary sequence turned to progradation, and the lithology shifted to yellow-green silty shale as the degree of water reducibility decreased. The calcitic content increased around the basin centre, indicating a shallow water depositional background (Figure 9). The water kept invading, submerged the uplifts in the north and channeled the Qianzhong and Xuefeng Uplift in the south. Only part of the northern margin was still of good organic preservation condition along these uplifts.

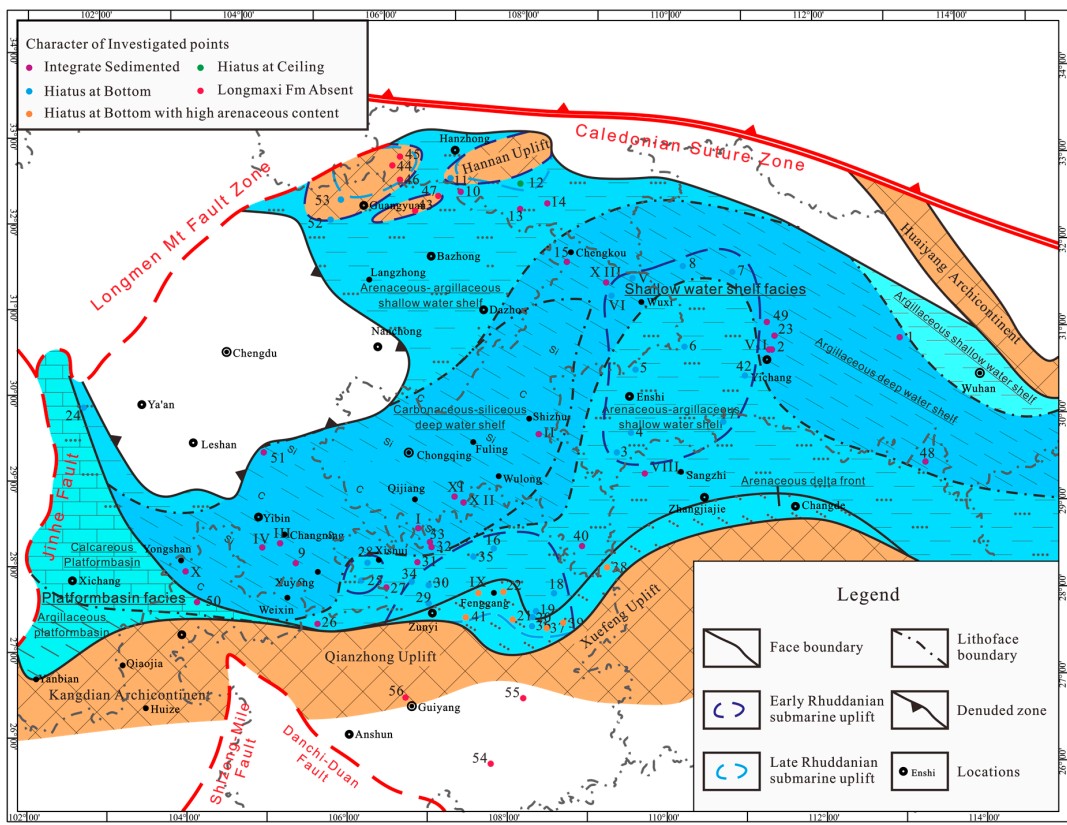

**Figure 8.** Sedimentary environment and lithology distribution of Rhuddanian in Middle–Upper Yangtze.

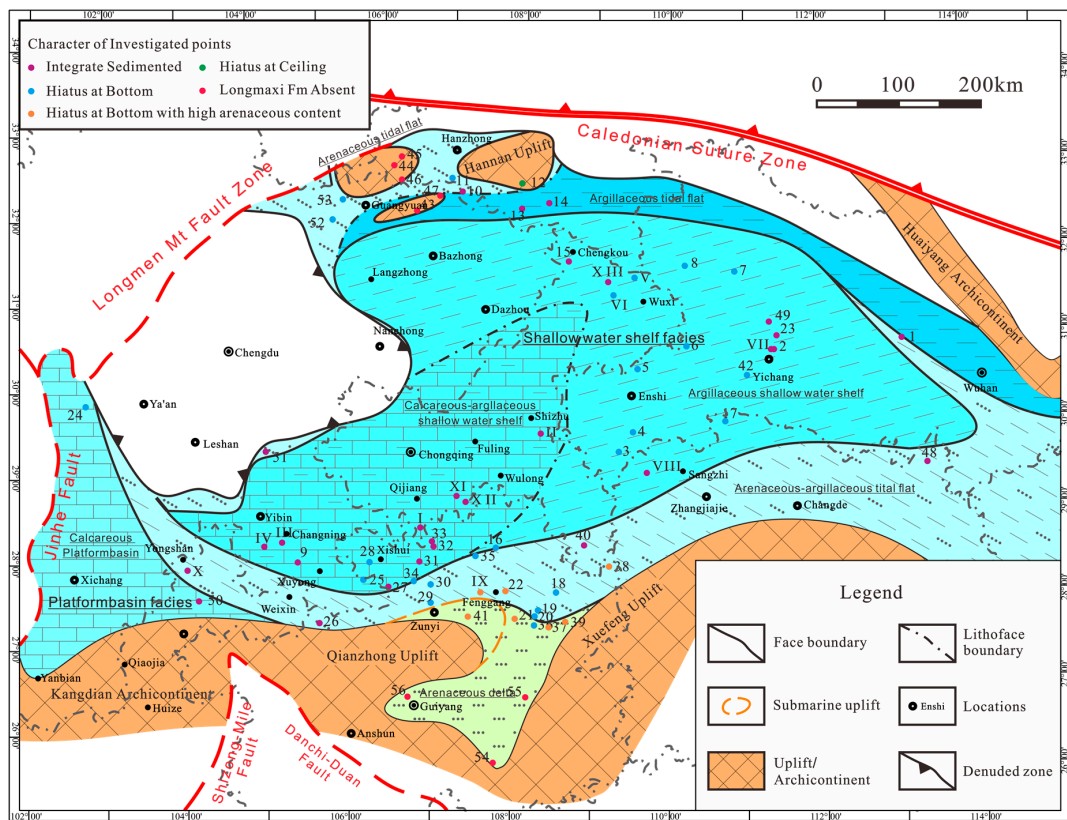

**Figure 9.** Sedimentary environment and lithological distribution of Aeronian in Middle–Upper Yangtze.

## 5. Impact of Sedimentary Environment on the Longmaxi Shale Reservoir

### 5.1. Coupling of Dark Shale Development and Water Background

Discussions on the development of organic-enriched shale indicate that it is influenced by comprehensive mechanisms, including sea level change, water condition (depth, reductivity), and sedimentary environment [6,42,43]; among them euxinic and reductive deep water are the key factors. The water was layered with lentic and reductive condition in the lower zone, and normal water for the living species in the upper layer. The maximum living depth of plankton, like graptolite, indicated the ceiling of the lower layer that was unsuitable for living. The plankton zonation discussed [44] showed that the living zone of graptolite deepened since *cyphus* of the late Rhuddanian (Figure 10), demonstrating the decline of the reductive water ceiling and the deterioration of conditions for organic matter preservation. Except that of the area influenced by uplifts, the ceiling of dark shale at the basin centre descended later than that near the uplifts, indicating longer duration of the environment that was advantageous for organic preservation. There was still organic-enriched shale that developed in the Aeronian over a large area in the basin centre, such as Point III.

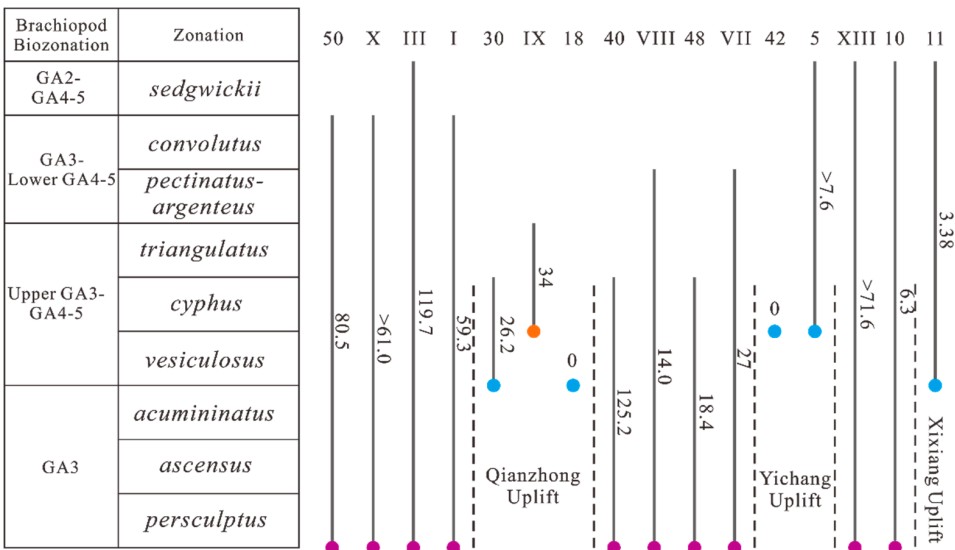

**Figure 10.** Comparison of dark shale development and fossil zonation (the number and meaning of point color is the same as Figure 2).

### 5.2. Sedimentary Environment Suitable for Dark Shale Development

The control of the sedimentary environment on the shale gas reservoir mainly includes mineral composition, lithological association, and texture. By statistical analysis of reservoir characteristics in different sedimentary environments (Table 1), the Longmaxi Formation showed a stable distribution of lithological associations, textures, and regular changes vertically. The lower member was the highest organic content sediment, with enriched graptolite fossils along the well-developed bedding, and the silica content and porosity were higher than the upper member. The upper shallow water sediment consisted of more arenaceous content with the bedding shown by compositional changes, or denser in the centre zone with high calcareous content.

The lower member of the Longmaxi Formation developed in euxinic deep-water, and it is of high TOC and quartz content. Large quantities of organic matter would form considerable micropore space during thermal evolution, meanwhile the abundance of graptolite fossil correlated well with TOC content, which could form sheet-like pores along the bedding (Figure 11a,b) [46]. The high silica content is related with biogenetic factors and volcanic activities [47,48]. The upper member, which formed under shallow-water shelf or tidal flat environments with a lower reduction degree is of relatively lower TOC content, and the calcareous or sandy laminae are alternately layered with

organic matter enriched ones (Figure 11c). Upper members forming closely around the uplifts kept a high brittleness and porosity, for example, at Point IX, the debris content near the ceiling of Longmaxi Formation can be 32%.

**Table 1.** Main sedimentary environment and corresponding reservoir characteristics.

| | Typical Profile | Lithofacies | Thickness (m) | TOC (%) | Brittle Index (%) | Porosity (%) |
|---|---|---|---|---|---|---|
| Deep water shelf | Lower Qijiang (I) | Carbonaceous-siliceous | 35.7/35.7 # | 0.99–4.77 * 2.99(6) | 53.2–61.3 (2) | 2.45–3.69 2.98(3) |
| | Lower Shuanghe (II) | Carbonaceous-siliceous | 49.5/49.5 | 2.00–5.35 3.38(28) | 36.6–80.1 58.9(12) | 1.76–9.66 6.44(8) |
| | Lower Yongshan (X) | Carbonaceous | >37.6/>37.6 | 0.94–2.52 1.59(16) | 37.3–77.9 58.2(16) | 2.21–10.81 7.73(3) |
| | Lower WX2 (XIII) | argillaceous | 77.4/77.4 | 2.13–8.00 3.47(15) | 64.0–76.0 70.8(9) | 0.53–2.80 1.26(4) |
| Shallow water shelf | Upper Qijiang (I) | Carbonaceous-argillaceous | 23.6/101.6 | 0.27–1.05 0.79(8) | 44.6–52.2 (2) | 1.54–6.47 4.43(3) |
| | Upper Shuanghe (II) | Carbonaceous-argillaceous | 70.2/>91.3 | 0.29–2.00 1.11(47) | 27.3–46.4 36.7(27) | 1.71–4.28 2.85(8) |
| | Upper WX2 (XIII) | Argillaceous-silty | >68.6/>71.4 | 0.06–1.52 0.38(8) | 66.0–67.0 66.3(3) | 0.7–3.47 1.50(8) |
| | YY1 [45] | Argillaceous | 19.3/523.1 | 0.98–3.47 2.00(15) | 37.0–66.0 50.3(8) | 1.6–2.0 1.7(8) |
| Tidal flat | Upper Fenggang (IX) | Arenaceous-argillaceous | 32/128 | 0.13–0.41 0.27(9) | 30.5–66.18 44.3(4) | 1.31–3.77 2.78(4) |
| | Upper Yongshan (X) | Arenaceous-argillaceous | 23.4/181.3 | 0.5–1.15 0.66(6) | 48.9–70.6 63.0(6) | 1.62–7.49 4.53(4) |

* min-max; ave (counts). # thickness of dark shale/total thickness of Longmaxi Formation.

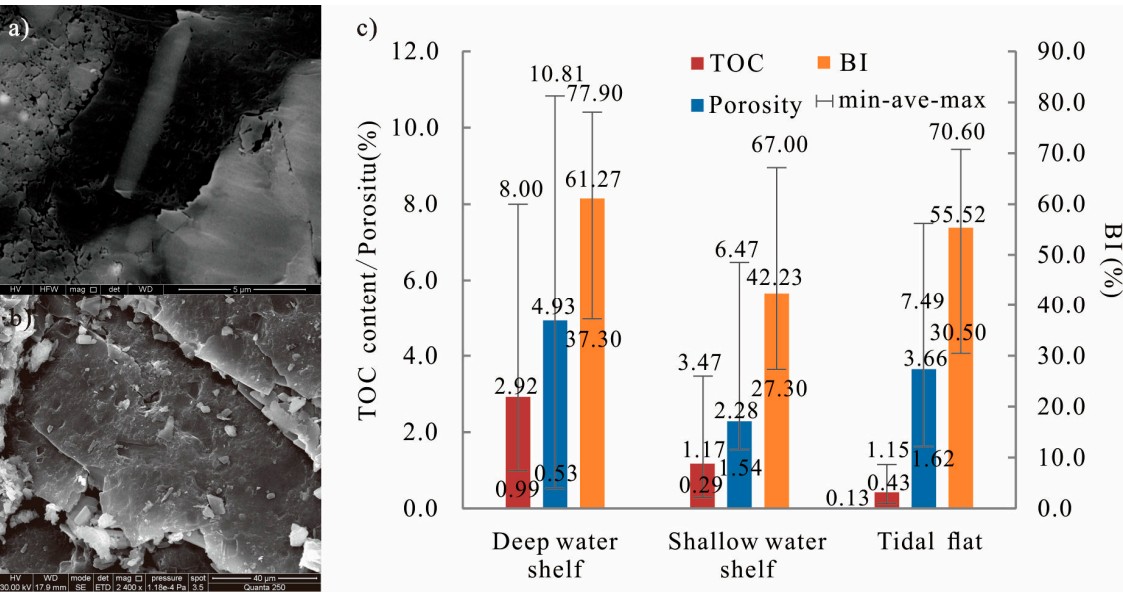

**Figure 11.** Reservoir pore characteristics, pores related with (**a**) organic matters and (**b**) carbonized graptolite, and (**c**) coupling of shale gas reservoir characteristics and sedimentary environments.

## 5.3. Reservoiring Association Characteristics of the Longmaxi Formation

By comprehensively analyzing the lithologies and their associations of the Longmaxi Formation in the Middle and Upper Yangtze, typical reservoiring models were built for four typical regions

(Figure 12). The continuous deposition regions, like Shuanghe (III) and Qijiang (I) profile, where the accumulation and seal system are well developed. The accumulation system, at the bottom of Longmaxi Formation, is high in TOC content and porosity, with a thickness of dark shale over 50 m for most of the region, and the dark shales can even be found in the last fossil zonation of Aeronian, *segdwikii*. Meanwhile, the low TOC content shale in the upper member is helpful in forming both lithological and hydrocarbon concentration seal systems. The overlying Shiniulan Formation or Luoreping Formation is a set of laterally continuous distributed, fine clastic sediment that can work as the regional seal cap so as to form a better seal system.

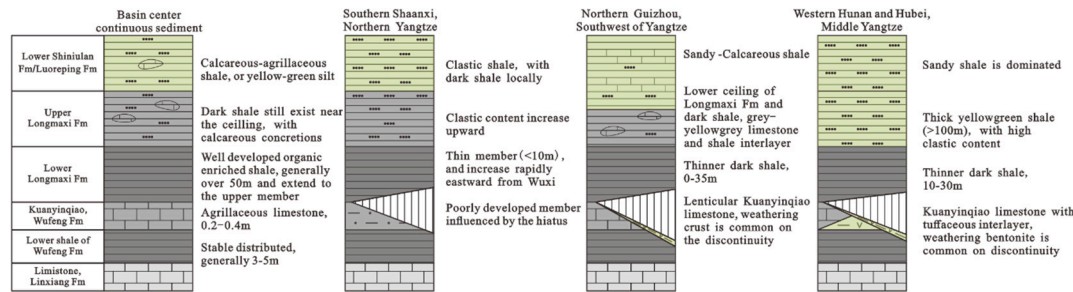

**Figure 12.** Reservoiring association characteristic of Longmaxi Formation.

The dark shale of Guangyuan–Nanjiang in the northern Yangtze is organic-rich matter but the thickness is rather small (<10 m), and the distribution is discontinuous, due to episodes of uplifts. Meanwhile, the discontinuity at the bottom of the Longmaxi Formation is common in the northern Yangtze. On the contact surface between the Longmaxi and underlying formations, yellow-green shale (Nanzheng Shale) or pebbly sandstone appear as the interface. The lithology and sedimentary development of Longmaxi ceilings varies; most of the caps are thick yellow-green shales, while some are eroded. Southward at Chengkou–Wuxi, the discontinuity is difficult to determine merely from lithologies. At this region the duration of the deep-water environment spans more fossil zonation so that there formed thick dark shale. Meantime the association of upper member is similar with that of the basin centre, the conditions are good for gas accumulation.

Under the influence of uplifts, the lower member of Longmaxi Formation in northern Guizhou is rather thinner (<35 m) than that of the basin centre. The upper clastic member is of low hydrocarbon generation and sealing capability. The abundant fossils along the bedding would greatly enhance the lateral permeability, and the existence of weak interfaces in shale, such as bedding or laminae, and would lead to permeability variance in different directions. Experiments conducted on Longmaxi shale indicated that the permeability along the bedding can be ten times higher than that perpendicular to the bedding under reservoir conditions [49,50]. Meanwhile, the weathering palaeocrust on the bottom can be the diffusion pathway underlying during the geological revolution as it is closely adjacent to the Longmaxi bottom which is the main hydrocarbon source rock and reservoir. At the northern part influenced by the Qianzhong Uplift, the discontinuity is similar with that of the Wuxi area, but the dark shale is thinner than that of northern Chongqing because of the short duration of the deep-water environment.

The Longmaxi Formation sedimentation on the east of Yichang Uplift in Middle Yangtze is continuous but the discontinuous contact still exists in Wufeng Formation, and the organic-enriched shale is thin as well. However, the upper silty and sandy overlying member is able to work as effectively as a sealing cap, and laterally connecting with the basin centre. Tight gas can be the main gas reservoir in Middle Yangtze.

## 6. Conclusions

(1) The analysis of the lithologies and fossil development of the Middle and Upper Yangtze reflects the dynamic sedimentary environment evolution of Longmaxi Formation. The wide-spread

deep-water shelf in the early Rhuddanian and suitable organic matter preservation conditions led to the laterally continuous distribution of dark shale in the lower member. At the Aeronian, the sedimentary environment turned to a shallow-water shelf, and a ceiling of reductive conditions descended, thus the upper member is mainly organic lean shale. Around the uplifts or the margin of the basin, the sedimentary environment consists of delta front and tidal flat, and the sediments are arenaceous.

(2) The comparison of shale reservoir characteristics formed under different sedimentary environments showed that the deep-water shelf environment, accompanied by euxinic and reductive backgrounds formed the laterally continuous, high TOC content shale with large thickness and high porosity. Shales formed under a shallow-water shelf environment were of low TOC and high argillaceous content, with lower brittleness and porosity. Strata that formed under a tidal flat environment were influenced by calcareous sediments remarkably, and the brittleness and porosity were medium among the environments involved.

(3) Influenced by the discontinuity between the Ordovician and Silurian, the lithological association varies in the Middle and Upper Yangtze. The continuous deposition or the ambiguous discontinuous contact zones, formed under the long-lasting deep-water shelf environment developed the thick organic enriched shales, along with an effective upper sealing system, is considered the optimal shale gas reservoir, and the effective reservoirs are mainly distributed at southern Sichuan–eastern Chongqing and northern Chongqing.

**Author Contributions:** X.Q. and Y.Z. designed the project; X.Q. and Y.W. the experiments and analyzed the data; X.Q. wrote the paper and Y.Z. corrected it. Y.W. and F.S. modified the formats.

**Funding:** This study was supported by the National Natural Science Foundation of China (41802183), the National Postdoctoral Program for Innovative Talents (BX201700282), the China Postdoctoral Science Foundation (2017M621870) and the National Science and Technology Major Project (2017ZX05035004-002), the Scientific Research Foundation of Key Laboratory of Coalbed Methane Resources and the Reservoir Formation Process Ministry of Education (China University of Mining and Technology) (No. 2019-008).

**Acknowledgments:** We are grateful to the editors and reviewers for their constructive comments and suggestions.

**Conflicts of Interest:** The authors declare no conflict of interest.

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
