# Peer review of "The Early Silurian Sedimentary Environment of Middle-Upper Yangtze: Lithological and Palaeontological Evidence and Impact on Shale Gas Reservoir"

_minerals, doi:10.3390/min9080494_

Round 1

Reviewer 1 Report

The manuscript has the potential for a great article, but in its present form, it is not suitable for publication.

-          The research topic does not fall into the scope and aims of Minerals journal;

-          The major flaw: there is no Materials and Methods section. The reader can not understand how this study was performed, since there is no sample description, no analytical methods, and also the results (if there are any) are not clearly presented;

-          The manuscript was not well prepared and verified prior submission: page 6 line 176 to page 7 line 187 – these paragraphs includes some text that it is not belonging here, maybe from an Author Guidelines or from previously submissions.

-          Some figures and tables have small text sizes, that it will be difficult to read in the printed version of the article.

Author Response

Point-by-point responses to the reviewers′comments:

We thank the editor and all three reviewers for the constructive comments and suggestions. All our responses to each comment were highlighted in red font.

To·reviewer#1:

1.       The research topic does not fall into the scope and aims of Minerals journal;

Response: The trigger of the project is the ambiguous boundary of the basin boundary. The high-quality reservoirs of Longmaxi Fm centred at the basin centre formed under deep water shelf, so there is less attention on the boundary. However, there are still investigations in the less-qualitied regions, such as northern Guizhou which is under the influence of Qianzhong Uplift, and the boundary here is controversial. Are these regions still promising, why, are they of the potentials in forming other hydrocarbon reservoirs, all these are noteworthy. 

At the same time, there are abundant and detailed studies about the graptolite and biostratigraphy in the region in last century. Most of the researchers come from the Nanjing Institute of Geology and Palaeontology, and they pay more attention on the identification of graptolites, evolution of graptolites, and chronostratigraphy. These provide precious evidences for not only what they discussed, but the landscape, environment, which is helpful on the engineering and the understanding of the overall evolution of the basin during early Silurian. So we summarized our own surveys, and collected almost all of the profiles reported around the basin to comprehensively analyse the dynamic evolution of sedimentary environment and further impact on reservoir characteristics.

We believe that this manuscript is appropriate for publication by the Minerals journal because it is directly related to the sedimentary environment and shale reservoir.

2.       The major flaw: there is no Materials and Methods section. The reader can not understand how this study was performed, since there is no sample description, no analytical methods, and also the results (if there are any) are not clearly presented;

Response: Thank you for your kind suggestion. Now the methodology part is well concluded in the paper.  There are 13 outcrops or drills we analysed, and 56 profiles investigated in the previous studies. The experiments are conventional methods analysing the reservoir characteristics and have now presented in the paper.

3.       The manuscript was not well prepared and verified prior submission: page 6 line 176 to page 7 line 187 – these paragraphs includes some text that it is not belonging here, maybe from an Author Guidelines or from previously submissions.

Response: Thank you for your consideration. Accordingly, we have made changes as required.

4.       Some figures and tables have small text sizes, that it will be difficult to read in the printed version of the article.

Response: According to the reviewer′s good instruction, we’ve replotted the figures to fit the formatting.

Reviewer 2 Report

1. General comments:

English language and style need to be improved. There are a lot of typos and grammar errors. The introduction part is too short, and previous studies are not introduced in details. The importance of the current study and the basic information of studied samples are lacking.

2. Specific comments:

1)     Line 38. The beginning of citation number is not from [1].

2)     Line 62-69. Can you list those references in Table 1? What do those two question marks mean? What is the connection between Table 1 and Figure 1? Why did you choose “north” and “south” profiles for sedimentary environment change? Any reasons?

3)     Line 92. Section 2.1?

4)     Line 93-100. Those statements are lack of citations.

5)     Line 115-122. Citations are lacking.

6)     Line 123. Section 2.2?

7)     Line 146-156. Citations are lacking.

8)     Line 160. Section 2.3?

9)     Line 176-187. What are these sentences?

10) Line 241. What do those numbers mean? What do those colored dots mean? Can you explain those abbreviations in column 1?

11) Line 151. The units are lacking in Table 2. What is “BI”? brittle index?

12) Line 268. Did you propose the reservoir models or summarize from previous studies? If the models are proposed, discussion and comparison between proposed models and previous models should be added in section 4.3. If the models are summarized from previous studies, the statement and the reason should be added in section 4.3.

13) Line 267-301. Citations are lacking in section 4.3.

Author Response

Point-by-point responses to the reviewers′comments:

We thank the editor and all three reviewers for the constructive comments and suggestions. All our responses to each comment were highlighted in red font.

To·reviewer#2:

1.         General comments; English language and style need to be improved. There are a lot of typos and grammar errors. The introduction part is too short, and previous studies are not introduced in details. The importance of the current study and the basic information of studied samples are lacking.

Response: Thank you for your kind suggestion. Now the introduction part has briefly introduced the reason why we write this paper and introduced the previously studies about sedimentary environment and graptolite zonation.

2.         Specific comments:

1)     Line 38. The beginning of citation number is not from [1].

Response: We have made changes as required.

2)     Line 62-69. Can you list those references in Table 1? What do those two question marks mean? What is the connection between Table 1 and Figure 1? Why did you choose “north” and “south” profiles for sedimentary environment change? Any reasons?

Response: According to the reviewer′s good instruction, we have made changes as required. Two profiles, the north (53-44-11-12-14-XIII-VII) and south (I-33-30-41-20-18-40-VIII-48), are selected to represent the graptolite development and lithological association in the four target regions to illustrate the sedimentary environment change (Fig 1). These two profiles are representative to elucidate the change of environment as the profiles walk along the margins of uplifts involved.

3)      Line 92. Section 2.1?

Response: We have made changes as required.

4)      Line 93-100. Those statements are lack of citations.

Response: Most of the lines that lack of references are in Section 3, which is discussing the graptolite zonation and lithologies under the influence of uplifts and sea level change. The discussion is based on the points we investigated as listed in the caption of Figure 1, and the discussion is highly concluded. If we listed all the references, the formatting would be messy, so we add the key investigation points instead. The readers would track back the references by the points in Figure 1, and it’s labelled in the captions. There are several missing references in the last section and we’ve added them.

5)     Line 115-122. Citations are lacking.

Response: We have made changes as required.

6)     Line 123. Section 2.2?

Response: We have made changes as required.

7)     Line 146-156. Citations are lacking.

Response: There are several missing references in this section and we’ve added them.

8)     Line 160. Section 2.3?

Response: We have made changes as required.

9)     Line 176-187. What are these sentences?

Response: We have made changes as required.

10) Line 241. What do those numbers mean? What do those colored dots mean? Can you explain those abbreviations in column 1?

Response:

11) Line 151. The units are lacking in Table 2. What is “BI”? brittle index?

Response: We have added units. In addition, BI is a brittle index.

12) Line 268. Did you propose the reservoir models or summarize from previous studies? If the models are proposed, discussion and comparison between proposed models and previous models should be added in section 4.3. If the models are summarized from previous studies, the statement and the reason should be added in section 4.3.

Response: Thank you for your consideration. The models proposed are based on the association of lithologies, reservoir characteristics, thickness of the four typical conditions discussed. The models still follow the basic rules of unconventional reservoirs, source and reservoir share the same layer with a sealing cap.  The different is that the models take the discontinuous interface and lateral diffusion into consideration.  How the phenomenon observed experimentally works on the basin-scale still worth discussion, but it shall follow the same rule micro- and macroscopically, only on the macroscopic scale, there are more factors affecting the reservoir accumulation, such as tectonic structures and evolution.

13) Line 267-301. Citations are lacking in section 4.3.

Response: We have made changes as required.

Reviewer 3 Report

Dear Author(s),

You are advised to take suggestions (marked on the MS) into consideration.

Author Response

Point-by-point responses to the reviewers′comments:

We thank the editor and all three reviewers for the constructive comments and suggestions. All our responses to each comment were highlighted in red font.

To·reviewer#3:

1.         Another proofread is highly recommended to correct punctuation and grammar errors eg. in Abstract: line 13, "stable distributed"; pg. 2, line 61 "Fossil and lithology", etc.

Response: We have made changes as required.

2.         The figures are not readable and inaccurate.

Response: Thank you for your consideration. We’ve replotted the figures to fit the formatting and changed the inaccurate words.

3.    Any evidences for the uplifts took place during active sedimentation.

Response: The variation of zonation itself are good evidences for the activities of the uplifts. The glaciers were melting in the time span of Longmaxi Fm sedimentation, meantime there were tectonic activities, the sea level change was the overall result of these 2 aspects. As the result of sea level change, the Yichang highland was exposed during Wufeng sedimentation and submerged in Longmaxi, while the region with missing strata in northern Guizhou kept expanding at the same time span, indicating the activation of Qianzhong Uplift. The Qianzhong Uplift is a long lasting archicontinent that has a deep impact on the sedimentation of Yangtze since Cambrian, and there are abundant studies about it. References discussing the uplifts are 19, 32, 36.

Round 2

Reviewer 1 Report

The editor can decide if the manuscript falls into the journal topic.

Author Response

Dear academic editor,

Thank you for your letter and for the your comments concerning our manuscript entitled “The Early Silurian sedimentary environment of Middle-Upper Yangtze: lithological and paleontological evidence and impact on a shale gas reservoir” (Ref: minerals-542632). Those comments are all valuable and very helpful for revising and improving our paper, as well as the important guiding significance to our researches. We have studied comments carefully and have made correction which we hope meet with approval. We tried our best to improve the manuscript and made some changes of the manuscript. The minor mistakes are corrected directly in the manuscript. Below you will find our point-by-point responses to your comments. If you have any question, please contact us without hesitate.

We appreciate for your warm work earnestly, and hope that the correction will meet with approval.

Sincerely yours,

Yanming Zhu

For the question raised in the manuscript, the answers are as follows:

The relation of TOC content and graptolite abundance.

Response: It’s our former result published in ref. 46.  The reason about the relation is that as consumers, the abundance of graptolite is a good indicator for the abundance of producers in the food chain.  And now it is stated in Section 1.  There are few reports about plankton in Longmaxi Formation, because the plankton was decomposed and hard to define.

Do unconventional reservoir need upper sealing.

Response: Unconventional reservoir is defined as a continuous reservoir sealed by pressure, and there is no clear boundary around the reservoir.  It is true in the early Silurian shale reservoirs as well.  The hydrocarbon will diffuse, as time goes to infinity, there will be less difference of hydrocarbon content in source rock and surroundings.  If there is a low permeability cap, it will postpone this procedure.

There is another source rock in early Cambrian in Upper Yangtze, which is also shale, and the hydrocarbon generation condition is even better.  However, there is no successful commercial gas flow in the shale.  The reason behind is still unclear, and for our study the gas emission takes a great role.  The lower and upper layer of the Cambrian shale is carbonaceous, which are good reservoirs because of the dissolve of calcareous content, especially the upper Sinian (right beneath the Cambrian shale).  That is why we believe that if there is a sealing cap in unconventional reservoir, the association is much better.

Good pore structure in shale gas reservoir.

Response: There is a delicate balance of pore structure in unconventional reservoir.  The porosity and permeability cannot be too small otherwise there will be no gas flow, neither too big otherwise the gas will diffuse.  Usually the porosity and permeability of Longmaxi are around 3-6% and 0.1-20μD, respectively.  Permeability are of a bigger range because it is also related with the sample texture and direction tested.  Pore size distribution (PSD) are important as well, for the shales usually the PSD is bimodal, pores in nanoscale (<50nm) and fissures (>1μm) are well developed. 

The difficult thing is to relate the results tested physically and observed morphologically.  However, there are interesting things found in SEM images.  Usually the fossils are mineralized, the graptolite fossils in Longmaxi Formation are preserved as carbonized thin layers.  And if zoom in, there are more layers in the graptolite, like lasagne.  It is hard to say how much porosity the graptolite fossil contributed, it is unique, and maybe another reason why the Silurian Longmaxi shale is better than the Cambrian shale.  It is not closely related to this paper, and the result has published elsewhere.

The language proficiency.

Response: There are a lot of modification about writing, we are glad and somehow pity to see that.  Even we write in English, our mind still hiding somewhere in mother language controlling.  It takes time to improve and we are thankful for your help.

Reviewer 3 Report

Dear author(s),

After the revision, the MS is now acceptable by my side.

I assume it is improved and will attract the attention of the relevant international community.

Author Response

(The authors gave the same response as above.)
